# Hierarchical on-surface synthesis and electronic structure of carbonyl-functionalized one- and two-dimensional covalent nanoarchitectures

Christian Steiner[1], Julian Gebhardt[2,†], Maximilian Ammon[1], Zechao Yang[1], Alexander Heidenreich[3], Natalie Hammer[3], Andreas Görling[2], Milan Kivala[3] & Sabine Maier[1]

The fabrication of nanostructures in a bottom-up approach from specific molecular precursors offers the opportunity to create tailored materials for applications in nanoelectronics. However, the formation of defect-free two-dimensional (2D) covalent networks remains a challenge, which makes it difficult to unveil their electronic structure. Here we report on the hierarchical on-surface synthesis of nearly defect-free 2D covalent architectures with carbonyl-functionalized pores on Au(111), which is investigated by low-temperature scanning tunnelling microscopy in combination with density functional theory calculations. The carbonyl-bridged triphenylamine precursors form six-membered macrocycles and one-dimensional (1D) chains as intermediates in an Ullmann-type coupling reaction that are subsequently interlinked to 2D networks. The electronic band gap is narrowed when going from the monomer to 1D and 2D surface-confined $\pi$-conjugated organic polymers comprising the same building block. The significant drop of the electronic gap from the monomer to the polymer confirms an efficient conjugation along the triphenylamine units within the nanostructures.

[1] Department of Physics, Friedrich-Alexander University Erlangen-Nürnberg, Erwin-Rommel-Strasse 1, 91058 Erlangen, Germany. [2] Department of Chemistry and Pharmacy, Friedrich-Alexander University Erlangen-Nürnberg, Egerlandstrasse 3, 91058 Erlangen, Germany. [3] Department of Chemistry and Pharmacy, Friedrich-Alexander University Erlangen-Nürnberg, Henkestrasse 42, 91054 Erlangen, Germany. † Present address: The Makineni Theoretical Laboratories, Department of Chemistry, University of Pennsylvania, Philadelphia, Pennsylvania 19104-6323, USA. Correspondence and requests for materials should be addressed to J.G. (email: jugeb@sas.upenn.edu) or to M.K. (email: milan.kivala@fau.de) or to S.M. (email: sabine.maier@fau.de).

O n-surface synthesis is a powerful tool to fabricate atomically defined carbon-based nanostructures in a bottom-up approach with prospective applications in molecular electronics, optoelectronics and sensor devices[1–4]. Thereby, the structural control is essential, because the electronic properties of these nanostructures sensitively depend on their geometry.

The use of specifically designed organic precursors provides for on-surface synthesis of atomically precise graphene nano-ribbons[5,6] with tunable band gap through heteroatom-doping or by modifying their lateral size and edge termination[7–13]. In contrast, the growth of long-range ordered two-dimensional (2D) porous graphene in ultra-high vacuum (UHV) is still a major challenge, despite the use of various design concepts and surface reactions[14–18]. The bond flexibility in the molecular building blocks and the irreversibility of the newly formed C–C bond prohibit an error correction during the network formation, which leads readily to equilibrium-disordered structures of limited size[19]. Therefore, electronic properties of bottom-up fabricated 2D networks remain experimentally widely unexplored[20], despite the exciting physical properties such as a half-metallic character[21] or band structures with Dirac-cone-like crossings and linear band dispersion predicted by density functional theory (DFT)[22–24]. In line with graphene nanoribbons, porous graphene exhibits a band gap that can be controlled by the pore size, density, functionalization and geometry[25]. Hence, porous graphene promises to combine the outstanding electronic properties of graphene with a tunable band gap, which makes it more suitable than pristine graphene for many electronic and optoelectronic applications requiring gapped semiconductor materials[6].

Mostly carbon- and nitrogen-containing[26,27] 2D porous polymers have been studied so far, to avoid competing interactions among the functional groups during the reaction. With suitable precursors, there is no inherent limitation, however, to extend the surface-assisted synthesis to more complex 2D polymers that feature functionalized pores[28,29]. The functional groups facing the pores lay the foundation to further tune the electronic properties of porous graphene networks. In addition, functionalized pores can be used to selectively bind molecules, or to track chemical reactions in a confined space for catalytic applications.

Here we present the on-surface synthesis and electronic structure of covalent macrocycles, one-dimensional (1D) chains and 2D porous networks based on Ullmann-type coupling reactions of carbonyl-bridged triphenylamines (CTPAs) with a tuned substitutional pattern on Au(111). In the hierarchical on-surface synthesis, intermediate macrocycles and 1D chains are initially fabricated that are interlinked thereafter to 2D networks of high structural quality. Scanning tunnelling spectroscopy (STS) experiments reveal a narrowing of the band gap when going from the monomer to 1D and 2D surface-confined π-conjugated organic polymers comprising the same building block. First-principles DFT calculations qualitatively reproduce the electronic structure and confirm the weak coupling of the planar polymer to the surface that leads to a quasi-free-standing electronic structure.

## Results

**Hierarchical synthesis**. We aimed for a two-step hierarchical synthesis of the 2D polymers to avoid a high defect density that occurs typically in the direct synthesis[15,16]. Figure 1a shows the chemical structure of the CTPA compounds **1**, **2** and **3**, which were used as monomeric species and for the Ullmann-type synthesis, respectively (for synthetic details,

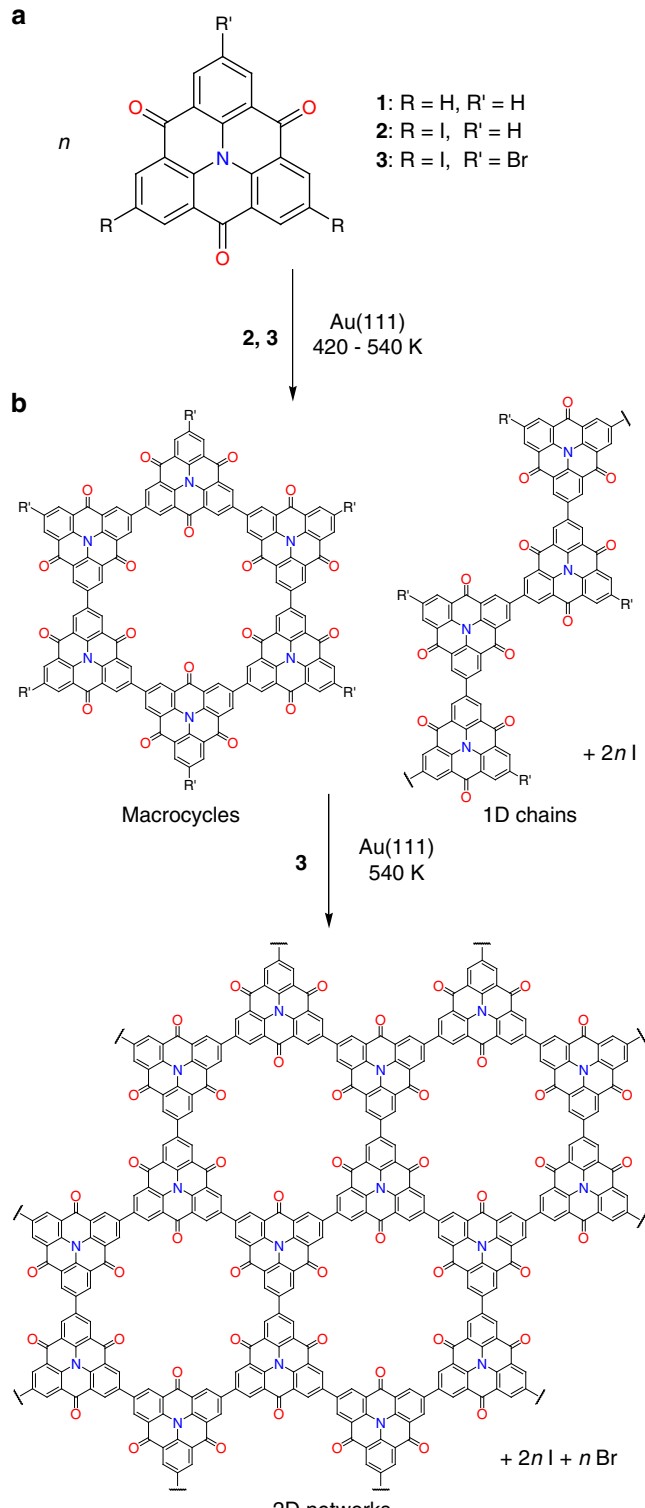

**Figure 1 | Hierarchical on-surface synthesis of 2D covalent polymers with carbonyl-functionalized pores.** (**a**) Structure of the used CTPA compounds **1**, **2** and **3**. (**b**) Proposed surface-assisted Ullmann-type reaction of CTPA towards macrocycles and 1D chains in the first step followed by their conversion into 2D networks in the second step.

see the Supplementary Methods). In this context, it should be emphasized that triphenylamines are widely applied as hole-transporting organic light-emitting diodes and field-effect

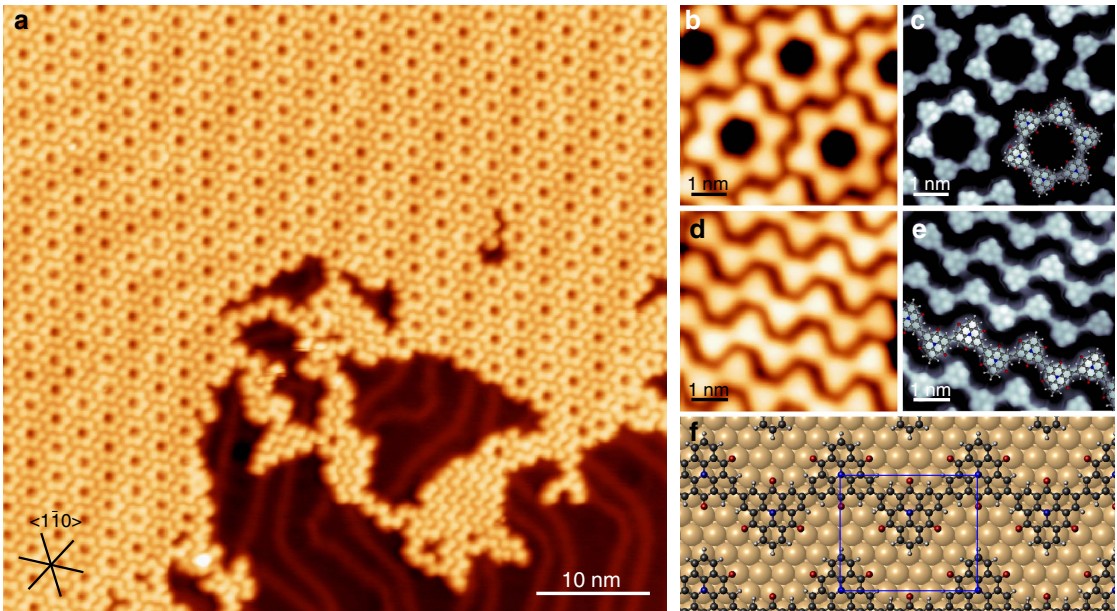

**Figure 2 | On-surface synthesis of nitrogen-doped chains and macrocycles.** (**a**) Overview STM image of the polymerized CTPA **2** after annealing the Au(111) surface to 540 K. (**b,d**) STM topographies of the covalently linked macrocycles and chains. (**c,e**) High-resolution STM images obtained with a functionalized tip revealing submolecular features; structural models are superimposed on the measurement (see also Supplementary Fig. 2). (**f**) DFT optimized structures (PBE + D3) of the polymerized CTPA chains on the Au(111) surface. STM Parameters: (**a**) $I = 360$ pA, $V = -160$ mV; (**b,d**) $T = 77.8$ K, $I = 50$ pA, $V = -250$ mV; (**c,e**) $T = 77.8$ K, $I = 1.3$ nA, $V = -50$ mV. High-symmetry directions of the Au(111) surface are indicated in **a**. Model in **c,e,f**: grey, white, red and blue spheres represent C, H, O and N atoms, respectively.

transistors[30,31]. It is known that the homolytic cleavage of the C–Br bond on Au(111) through gradual annealing takes place at higher temperatures than the cleavage of the C–I bond[16]. Therefore, the halogen-substituted CTPA precursor **3** initially forms covalent macrocycles and 1D chains in an intermediate step, and subsequently connect to a porous 2D polymer as a final product according to the reaction scheme in Fig. 1b.

**Synthesis of nitrogen-doped macrocycles and 1D chains.** First, we discuss the structure of the bottom-up fabricated nitrogen-doped macrocycles and 1D zig-zag-shaped chains. We used the diiodo-substituted CTPA **2** to suppress the second reaction step towards 2D structures in the on-surface polymerization. Scanning tunnelling microscopy (STM) topographies (Fig. 2a) reveal extended layers of self-assembled macrocycles and chains after sublimation of CTPA **2** at submonolayer coverage onto an atomically flat Au(111) that was held at room temperature and subsequently annealed to 540 K, to activate the surface-promoted aryl–aryl coupling (Fig. 1b). There is no selectivity towards the formation of either macrocycles or chains. On average, we obtain a yield of macrocycles of >20%, as determined from large-scale STM images. The mobility of the macrocycles and chains at the reaction temperature is sufficient so that they can self-assemble in extended long-range ordered islands with sizes of up to $80 \times 80$ nm² for the macrocycles. We note that the macrocycles and chains separate into individual domains, due to their dissimilar geometrical shape. In addition, the carbonyl groups promote highly ordered self-assemblies owing to H-bond formation. The dense-packed self-assembly of macrocycles (Fig. 2b) features a rhombic unit cell ($a = 2.91 \pm 0.07$ nm, $b = 2.91 \pm 0.07$ nm, $\theta = 60 \pm 1°$) and is stabilized by hydrogen bonds between the carbonyls and the hydrogens of the neighbouring phenyls (Supplementary Fig. 1a,b). The parallel chains (Fig. 2d) align such that the H-bond distance on both sides

of a CTPA unit is different. The unit cell comprising two CTPA units measures $a = 1.11 \pm 0.07$ nm, $b = 1.75 \pm 0.07$ nm, $\theta = 88° \pm 2°$ (see Supplementary Fig. 1c–e).

The CTPA molecules adopt a triangular shape in the STM images with no significant height variation, which indicates a planar adsorption. Perdew–Burke–Ernzerhof (PBE) + D3 calculations (see Methods) on the gold surface (Fig. 2f) confirm an almost planar adsorption with an averaged adsorption height of 3.26 Å. We used a molecularly functionalized tip to resolve intramolecular features (Fig. 2c,e and Supplementary Fig. 2) and to unravel the chemical structure of the reaction products. The centre-to-centre distance between the triangular shaped CTPA units in both, chains and macrocycles, extracted from these high-resolution images measures $1.01 \pm 0.03$ nm. This value is in good agreement with the centre-to-centre distance of 1.00 nm in our calculations for the 1D chains in vacuum at PBE + D3 level, confirming a successful covalent reaction.

The polymerization of the twofold symmetric CTPA precursor **2** almost exclusively leads to six-membered macrocycles and zig-zag-shaped chains. The geometrical stability of the hexagonal macrocycles results from the non-covalent H-bonds and van der Waals interaction forces between the macrocycles, which allows them to adapt their equilibrium state[19] and prevents the formation of pentagonal and heptagonal rings. Previously synthesized macrocycles on Ag(111)[32] and Cu(111)[33] were stabilized by an organometallic intermediate state that acts as a supramolecular template. We do not find any indication of an organometallic intermediate state in our experiments on Au(111) and hence we conclude that an organometallic template is no prerequisite for the geometrical stability of covalent six-membered macrocycles. Having shown the successful synthesis of six-membered macrocycles and chains, we address in the following their interlinking to 2D polymers. The hierarchical approach using these precursors is expected to

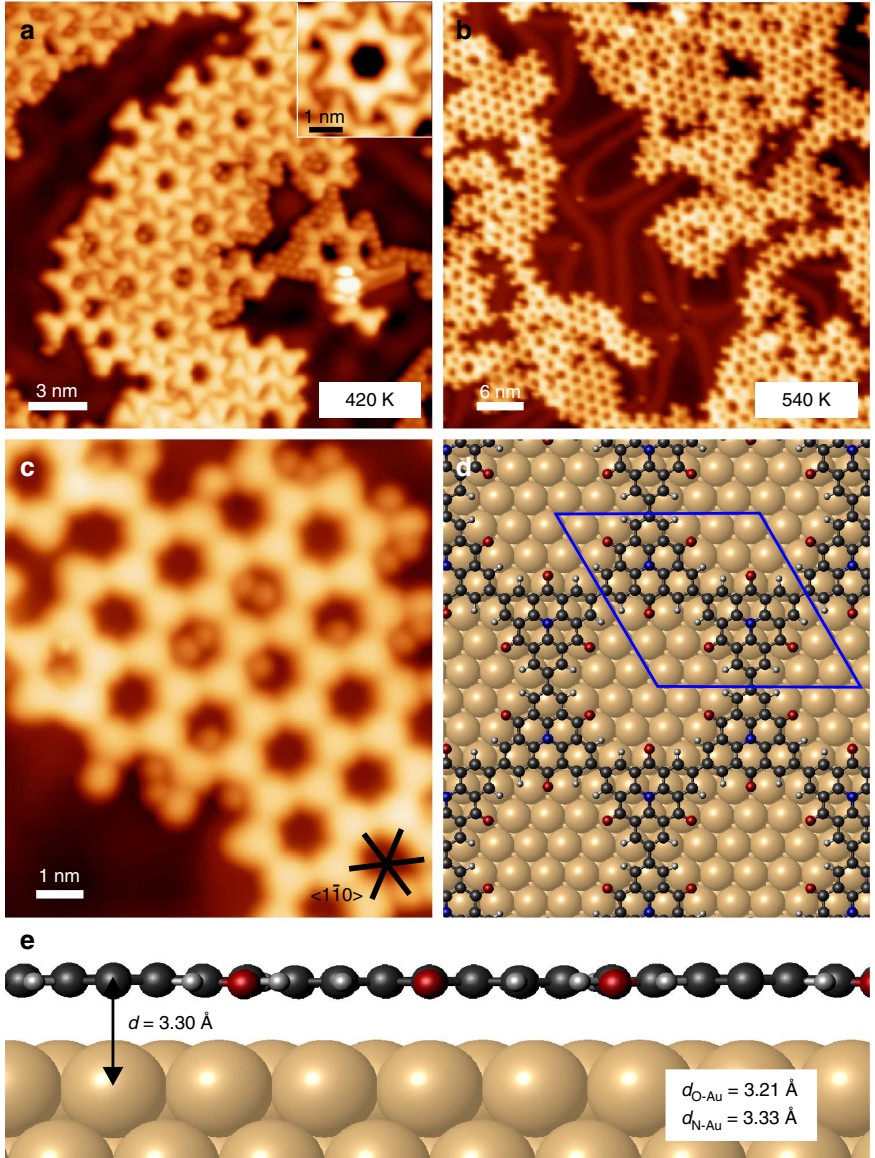

**Figure 3 | Hierarchical on-surface synthesis of 2D polymers with carbonyl-functionalized pores.** (**a**) Macrocycles and chains formed after the first reaction step at 420 K (deposition at 365 K). The remaining bromine atoms at the corners can be clearly identified in the STM images. (**b,c**) Covalently linked honeycomb network formed after the second reaction step at 540 K. (**d,e**) Corresponding DFT structures of the CTPA network on the Au(111) surface. STM parameters: (**a**) $I = 100$ pA, $V = -80$ mV, inset: $I = 800$ pA, $V = -80$ mV; (**b**) $I = 50$ pA, $V = -1$ V; (**c**) $I = 400$ pA, $V = -50$ mV.

reduce the defect density that still remains a difficulty in the synthesis of 2D polymers in UHV.

**Synthesis of 2D polymers with carbonyl-functionalized pores.** For the synthesis of the porous 2D polymer we used CTPA **3**, which has a bromine atom as third functional group. After the iodine cleavage, the polymerization (onset 390 K) into six-membered macrocycles and short chains (Fig. 3a) proceeds analogously as discussed above for diiodinated CTPA **2**. At this temperature, the bromines are still attached to the CTPA scaffold, as evident from the sharper edges in the STM images of both structures (inset in Fig. 3a) compared with Fig. 2b,d. The self-assembly of the Br-terminated macrocycles and chains is mediated through H-bonds between the carbonyls and C–H groups, as well as between Br and C–H groups in conjunction with halogen bonds between the Br and carbonyls.

Islands of porous graphene networks were formed in a second coupling step after the C–Br cleavage was induced by post-annealing the substrate to 540 K (Fig. 3b). The well-ordered networks (Fig. 3c) show a clear preference of six-membered pores with a few five- and seven-membered ones occurring at the periphery of the network patches. The suppression of defects within the porous networks is a major advantage of this hierarchical approach and is attributed to the preceding self-assembly of the macrocycles and chains, which allows undesired structural elements to segregate. The covalent linkage of the well-ordered six-membered macrocycles leads by design to the formation of honeycomb networks: the only possible defect would be a combination of four to eight-membered ring pairs, which is, however, not observed (Supplementary Fig. 3). The absence of these defects is most likely a consequence of their instability due to the large strain resulting from the inherent rigidity of the CTPA moieties. Pentagonal and

heptagonal defects at the periphery of the porous graphene islands may be created by interlinking short chains that are present at lower reaction temperatures before the second reaction occurs in the hierarchical approach.

In Fig. 3d–e, the PBE + D3 optimized geometry of the porous 2D polymer on Au(111) is shown. The optimized polymeric structure in vacuum has a lattice constant of 17.37 Å. Thus, the structure is commensurable to an optimized $(6 \times 6)$ Au(111) surface cell (the computed lattice constant $a_0$(Au) is 2.90 Å, in good agreement with the experimental value[34] of 2.88 Å). The CTPA polymer adsorbs almost planar on top of the gold surface (Fig. 3e) proofing the 2D character of the polymer. The largest distance in $z$-direction to the surface is found at the central N atom (3.33 Å), whereas the O atoms are closest to the surface (3.21 Å). The mean adsorption height of 3.30 Å indicates that the CTPA polymer is physisorbed on the Au(111). The adsorption energy is 3.55 eV per monomeric unit, 70% of which are attributed to dispersive (or van der Waals) forces, supporting the physisorptive character of the interaction. The physisorption of the CTPA polymer is also confirmed by a charge density difference (CDD) analysis.

**Electronic structure of the chains and porous network.** Structural control with atomic precision is essential for studying the electronic structure of porous graphene networks, because their band gap sensitively depends on the specific atomic configuration. The approach presented here yields 2D and 1D polymers built from identical building blocks with satisfactory structural quality. Notably, the direct comparison between the electronic structure of bottom-up fabricated 2D polymers and their analogous 1D counterparts has so far been explored theoretically for extended $\pi$-systems[22,35], but has, to our knowledge, not been experimentally verified.

In our STS measurements, we excluded networks with trapped halogens in the pores to avoid doping effects[36]. In addition, networks with a size less than seven rings and chains shorter than eight units were neglected to exclude an interfering size dependence[35] in the measured electronic properties and to ensure maximum reproducibility. The measured peak energies in STS were found to be independent of the tip-molecule distance, indicating that there was no observable effect of the electric field in the tunnel junction (Supplementary Fig. 4). First, we discuss the band gap based on differential conductance (d$I$/d$V$) spectra shown in Fig. 4, recorded at different sites on the 2D covalently linked network resulting from CTPA **3**, the 1D chains that are fabricated from CTPA **2** and the hydrogen terminated CTPA **1** (ref. 37) monomer (see Supplementary Fig. 5). The 1D chains are found as individual chains and self-assembled in 2D H-bonded islands, which will be discussed separately.

The valence band edge (VBE) appears as a clear shoulder above the Au reference below the Fermi level. In contrast, the conduction band edge (CBE) inclines likewise gently to a prominent CB peak above the Fermi level. The specified energies are averaged values over representative spectra, measured apart from defects or irregular structures. We observed the VBE for all three structures at similar energies: $-1.63 \pm 0.04$ eV (2D network), $-1.68 \pm 0.03$ eV (self-assembled chains) and $-1.64 \pm 0.03$ eV (isolated chains). In contrast, we find that the energy of the CB resonance scales with the dimensionality: $1.51 \pm 0.03$ eV for the 2D network, $1.50 \pm 0.06$ eV and $1.67 \pm 0.03$ eV for the self-assembled and isolated chains, respectively. The CB resonances are slightly asymmetric indicating that they do not represent a single band, instead contain some unresolved states (see also discussion of the band structures below). The shift in the CB energy from isolated to

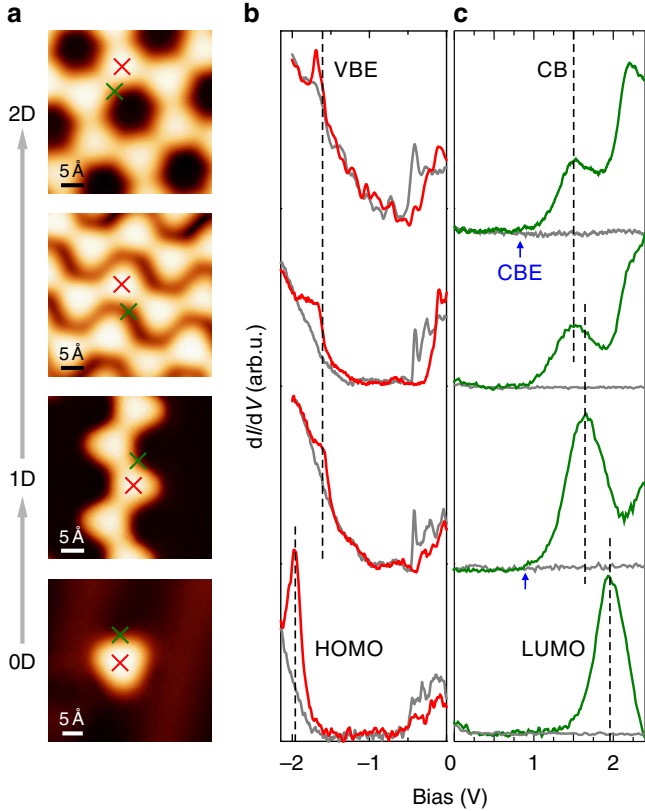

**Figure 4 | d$I$/d$V$ spectra measured on the 2D and 1D CTPA polymer and CTPA monomer.** (**a**) STM images indicating the location of the STS spectra. (**b,c**) STS of the VBE (**b**) and the CB (**c**) of the 2D network (top), self-assembled 1D chains (second row) and isolated 1D chains (third row), and the corresponding HOMO and LUMO of the monomer CTPA **1** (bottom). The grey spectra were taken on the bare Au(111) surface as a reference. The blue arrows indicate the onset of the CBE. STM parameters: (**a**, 2D) $I = 50$ pA, $V = -50$ mV; (**a**, self-assembly) $I = 180$ pA, $V = 100$ mV; (**a**, 1D, 0D) $I = 50$ pA, $V = 1$ V. Open feedback parameters: $V_{mod} = 12$ mV$_{rms}$; (**b**, 2D) $I = 700$ pA, $V = -4$ V, (**b**, self-assembly, 1D, 0D) $I = 100$ pA, $V = 1$ V; (**c**, 2D) $I = 700$ pA, $V = -4$ V; (**c**, self-assembly, 1D, 0D) $I = 50$ pA, $V = 1$ V.

self-assembled chains provides evidence for electrostatic effects or weak electronic hybridization due to the H-bonds among the chains. This is in line with previously reported shifts in molecular energy levels of self-assembled $\pi$-conjugated molecules due to polarization effects by the local electrostatic environment or small differences in the H-bond lengths[38,39]. We attribute the broader scattering of the CB energy in the self-assembled chains to different H-bond lengths in the self-assembly. The influence of the H-bonds on the CB energy is also validated by chains at the edge of the self-assembly, where we measure $1.61 \pm 0.03$ eV for the CB, which is in between the isolated and the self-assembled chains, see Supplementary Fig. 5c. A similar scaling of the energy shift with the dimensionality is also observed for higher unfilled states, the CB + 2 (see Supplementary Fig. 5): $2.22 \pm 0.03$ eV for the 2D network, $2.21 \pm 0.06$ eV for self-assembled and $2.42 \pm 0.03$ eV for isolated chains. Finally, we compare the measured band energies with the highest occupied molecular orbital (HOMO) and lowest unoccupied molecular orbital (LUMO) of the hydrogen substituted CTPA **1** monomer, which can be considered as the 'zero-dimensional' (0D) equivalent to the 1D and 2D polymer structures. The hydrogen substitution ensures no additional shift of the frontier orbitals due

to the functional groups. STS spectra were recorded on monomers evaporated on Au(111) held at 80 K. The pronounced peaks at $-1.96 \pm 0.03$ eV and at $1.95 \pm 0.03$ eV, respectively, are attributed to the HOMO and LUMO of the CTPA **1**.

In conclusion, the measured gap between the VBE and the first CB resonance scales with the dimensionality of the covalent structures: $3.14 \pm 0.04$ eV (2D networks), $3.18 \pm 0.06$ eV (self-assembled chains) and $3.31 \pm 0.03$ eV (isolated chains). The measured peak energies of the CB resonances instead of the CBEs provide indirect evidence for the band gap scaling. However, the main features of the CB are comparable in the calculated density of states (DoS) for 1D and 2D structures, and therefore we expect the energy gaps VBE–CB and VBE–CBE to scale similar among the dimensions. For the 2D network and isolated chains, the approximate CBE onset appears at $0.82 \pm 0.05$ eV and $0.86 \pm 0.06$ eV, respectively, in the STS spectra (blue arrow), which results in an approximate band gap of $2.45 \pm 0.09$ eV and $2.50 \pm 0.09$ eV, respectively. In contrast, the HOMO–LUMO gap of $3.91 \pm 0.03$ eV measured for the CTPA **1** monomer is significantly larger. We can conclude from the STS measurements that the band gap decreases around 170 meV from 1D to 2D structures built from CTPA compounds and is mainly caused by a shift of the CB, whereas from the monomer to 1D chain the gap decreases by around 1.34 eV. The decrease of the band gap from the monomer to the 2D polymer is a result of the increased effective $\pi$-conjugation length, which indicates an efficient conjugation among the triphenylamine building blocks within the nanostructure. The small difference between the band gaps of 1D and 2D compared with 0D and 1D shows that the $\pi$-electron delocalization length is only little increased between 1D and 2D CTPA polymers.

The electronic Shockley surface state of the Au(111) substrate is observed by the steep rise in the STS spectra near $-0.5$ eV below the Fermi level (Fig. 4b and Supplementary Fig. 6). The surface state persists in the d$I$/d$V$ spectra above the porous 2D polymer, which indicates a weak confinement of the surface state by the covalent pores. It shifts around 130–250 meV towards the Fermi level due to the modification of the work function caused by the molecular adlayer. The surface state measured in STS maps in the centre of the pore is weakly confined by the molecular network (Supplementary Fig. 7), which is in accordance with the physisorbed nature of the network.

Constant height d$I$/d$V$ maps at energies of the observed VBE and CB states provide further insights into the localization of the states. The DoS is localized at equivalent positions in both the covalently linked 2D network and 1D chains (see Fig. 5). At the VBE (Fig. 5b,f), the homogeneously distributed DoS at the CTPA centres suggest a homogeneous electron density across the monomer backbone, whereas extended delocalization of the $\pi$-electrons along the $sp^2$-carbon framework between CTPA units is questionable due to a decreased relative intensity at the new C–C bonds. The DoS of the first and second resolved unoccupied band is localized beside (Fig. 5c,g) and on top (Fig. 5d,h) of the newly formed intermolecular C–C bond, respectively, with weakened intensity at the CTPA centres. The presence of unoccupied states at the positions of the newly formed C–C bond has also been observed in previous STS experiments on covalent networks[40]. The DoS of the CTPA **1** monomer (Fig. 5j,k) matches well with the 1D and 2D structures: the HOMO is resolved in the centre of the molecule and the LUMO of the monomer is located next to the carbonyl bridges at the periphery of the carbon scaffold. This suggests that the LUMOs of two neighbouring CTPA monomers hybridize upon the covalent connection, which form the unoccupied state located beside the

C–C bonds in the polymer. Qualitatively, no difference in the DoS has been observed in constant current and constant height d$I$/d$V$ maps in accordance with the planar adsorption geometry of the polymer structures on the surface (see Supplementary Fig. 8). The bias dependence of the STM contrast reflects as well the observed features in the STS maps (Supplementary Fig. 9). We note that the location of the first unoccupied band beside the newly formed C–C bond gives the pore an exceptional electronic structure suitable for host-guest interactions, an aspect which will be published elsewhere.

We now discuss our DFT results in order to get more insights into the electronic structure. Figure 6 shows the calculated band structure for both the self-assembled 1D chains and the 2D network in vacuum, for further assignment of atomic species to states the projected DoS is shown. In the 2D network, most bands show little to no dispersion (see Fig. 6e–f and Supplementary Fig. 10), indicating only a small level of band hybridization. Both VBE and CBE are predominately hybridized states of carbon and nitrogen $p_z$, with some admixture of oxygen density. The VBE is located at the $\Gamma$ point, whereas the CB is quasi dispersionless, making the 2D CTPA effectively a semiconductor with a direct band gap of 1.74 eV.

The experimentally observed band gap increase upon decreasing the dimensionality is nicely reproduced at the DFT level. The PBE + D3 computed band gap increases to 1.84 and 1.86 eV for self-assembled and isolated 1D chains, respectively, and further to a HOMO–LUMO gap of 2.27 eV in the case of the 0D monomer **1**. We note that shifts of the frontier orbitals owing to the halogen groups are avoided by the H-substitution. In comparison, the HOMO–LUMO gap of monomer **2** is 2.03 eV. The effect of the reduced dimensionality is visualized by comparing the band structures (Fig. 6b,e) of the two systems. In the 1D case (Fig. 6b), the band dispersion of the interacting $\pi$-bands with carbon and nitrogen character around the Fermi level is decreased along $\overline{\text{KY}}$ compared with the 2D case ($\overline{\text{KM}}$ in Fig. 6e). This is observed in the VB and (to a smaller extent) in the CB, leading to an overall decrease of the band gap by 0.1 eV. In addition, the dispersionless band of the CBE is missing in the 1D case (for a more in depth analysis, see Supplementary Discussion).

Because of the systematic qualitative underestimation of band gaps computed by the general gradient approximation, we compare our PBE + D3 results with results obtained with the functional due to Heyd, Scuseria and Ernzerhof (HSE) (Supplementary Fig. 11), which is known to yield quantitatively more accurate band gaps. The band structure for the 2D case is displayed in Fig. 6f, the 1D structures in Fig. 6c. Overall, the band structure is mainly retained; however, the dispersionless bands with oxygen contribution that are close to the VBE in the PBE + D3 results are lowered in energy. In addition, the band gap is significantly increased by $\sim 0.7$ eV. These changes are also observed in the 1D cases and the band gaps for 2D to 0D systems are compared for both methods with our experimental results in Table 1. As expected, the HSE band gaps are closer to the experimental values, whereas the relative trends of structures with different dimensionality are similar for both computational models, although the gap change with the dimensionality of the system is slightly increased. With the hybrid functional, the calculated results are in good agreement with the STS experiments.

Next, we discuss the influence of the Au(111) substrate. As PBE + D3 and HSE calculations yield qualitatively identical electronic structures for the free-standing cases, we choose the computationally more affordable method for the systems that include the substrate. DoS and band structure of the surface-supported 2D polymer are shown in Fig. 7. The overall appearance remains unchanged compared with Fig. 6d,e, that

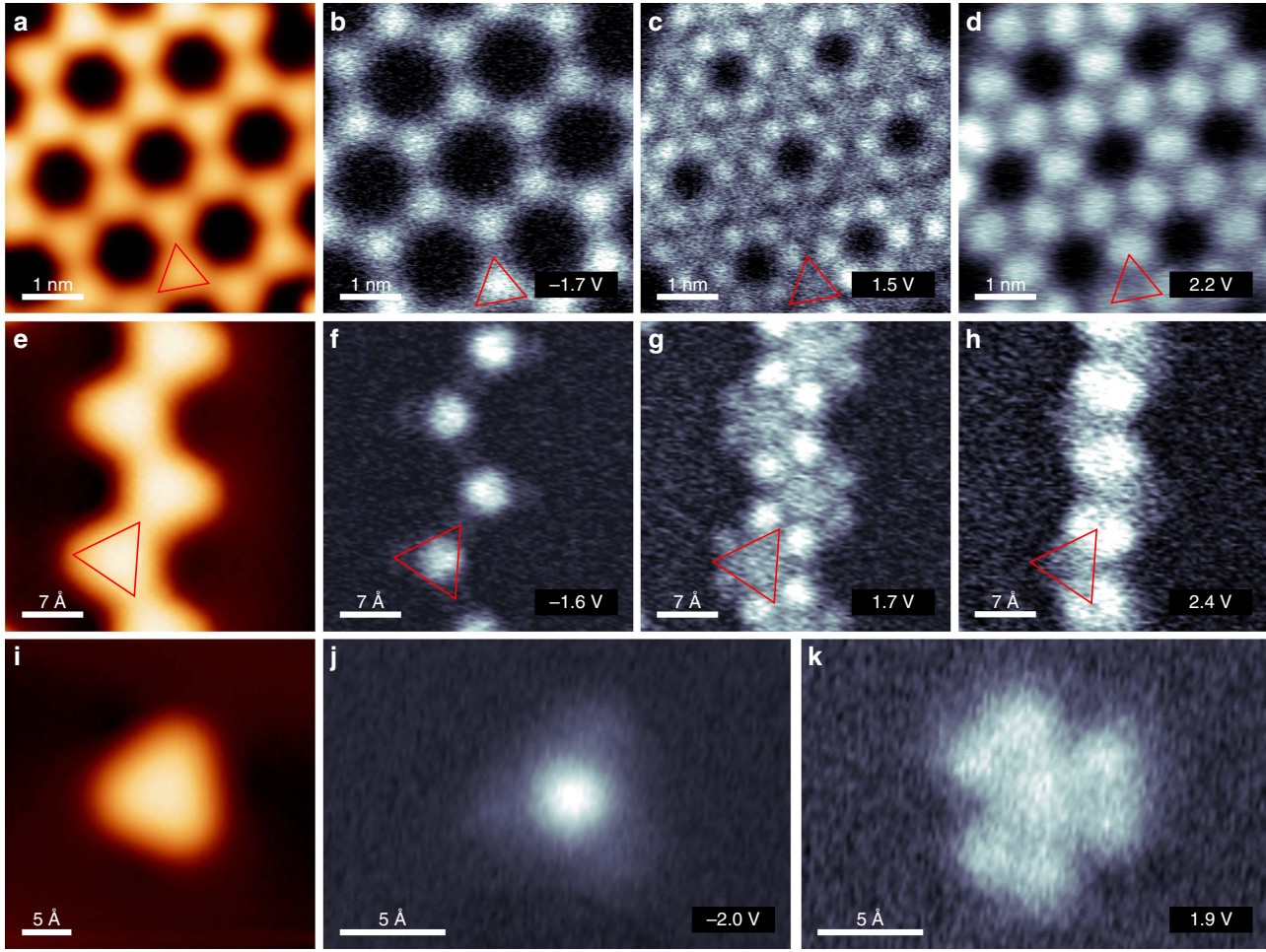

**Figure 5 | d*I*/d*V* maps of the 2D and 1D CTPA polymer and CTPA monomer.** Topography and constant height d*I*/d*V* maps at bias voltages corresponding to the VBE, the CB and CB + 2 of the 2D polymer (**a–d**) and isolated chain (**e–h**). The topography and corresponding constant height d*I*/d*V* maps at the HOMO and LUMO of the monomer CTPA **1** are presented in **i–k**. The red triangles depict the topographic shape of one CTPA unit. STM parameters: (**a**) *I* = 50 pA, *V* = − 50 mV; (**e**) *I* = 50 pA, *V* = 1 V; (**i**) *I* = 180 pA, *V* = − 0.5 V. Open feedback parameters: *I*$_{set}$ = 180 pA and *V*$_{mod}$ = 12 mV$_{rms}$.

is, the $p_z$ and $p_x, p_y$ contributions are well separated, being dominated by oxygen, as well as nitrogen and carbon around the Fermi level, respectively. This similarity of adsorbed and gas-phase structures demonstrates the weak interaction and thus the physisorption of CTPA on Au(111). The weak adsorbate substrate interaction is further demonstrated by a CDD analysis (Supplementary Fig. 12). Owing to the large adsorption distance above 3 Å, the effect of the electrostatic surface potential and the pillow effect[41] are small and in perfect agreement with the interactions observed for graphene and metal surfaces[42]. For the latter, chemisorption is observed at smaller adsorption distances (∼2 Å) for a strongly interacting surface, such as Ni(111), with charge density rearrangements an order of magnitude larger than in the physisorbed case, which is observed on weakly interacting surfaces, for example, Au(111) or via intercalation[43,44]. Nevertheless, the weak interaction with the substrate leads to a small charge transfer (0.09 excess electrons are attributed to each CTPA unit by a Bader analysis)[45], shifting the adsorbate bands ∼1 eV towards larger binding energies. In addition, hybridizations with many gold bands are observed, leading to numerous avoided crossings and a number of subtle changes. The valence region remains mostly unchanged, allowing to retrace the adsorbate band dispersion, with the dispersionless O$_{px,py}$ bands, as well as the dispersion of the VB with mainly C, N$_{pz}$ character. Owing to avoided crossings, however, the DoS of

the VB is slightly changed, showing two peaks compared with the broad distribution in the gas phase. More changes are visible in the low-lying CBs. Especially the CBE around Γ is lowered, leading to a reduction of the band gap in the case of the 2D adsorbate by 0.17 eV. In addition, the CB + 1, which appears as separated peak around 1.9 eV in the gas phase, is broadened and shifted downwards towards the CB. Consequently, these two bands are not resolvable in the STS measurements and are observed as one broad state (Fig. 4).

Figure 7c shows an analogous band structure of a self-assembled 1D CTPA chain on Au(111). Owing to the weak interaction of the physisorbed CTPA, the effects of the surface are similar to the 2D case, that is, despite hybridization with many substrate states, the bands of the adsorbate can be retraced and a small decrease of the band gap by 0.06 eV is observed. Thus, the effects of reduced dimensionality discussed above, such as the band gap decrease, are transferrable to the substrate supported case.

Figure 7d shows the calculated occupied and unoccupied partial charge densities for bands around the Fermi level in the 2D network adsorbed on Au(111). The VB shows density distributed on the full backbone of the CTPA and the nitrogen centres. This explains the bright feature observed experimentally in the d*I*/d*V* map of the VBE in Fig. 5b. The density is only delocalized over the inner carbon rings of

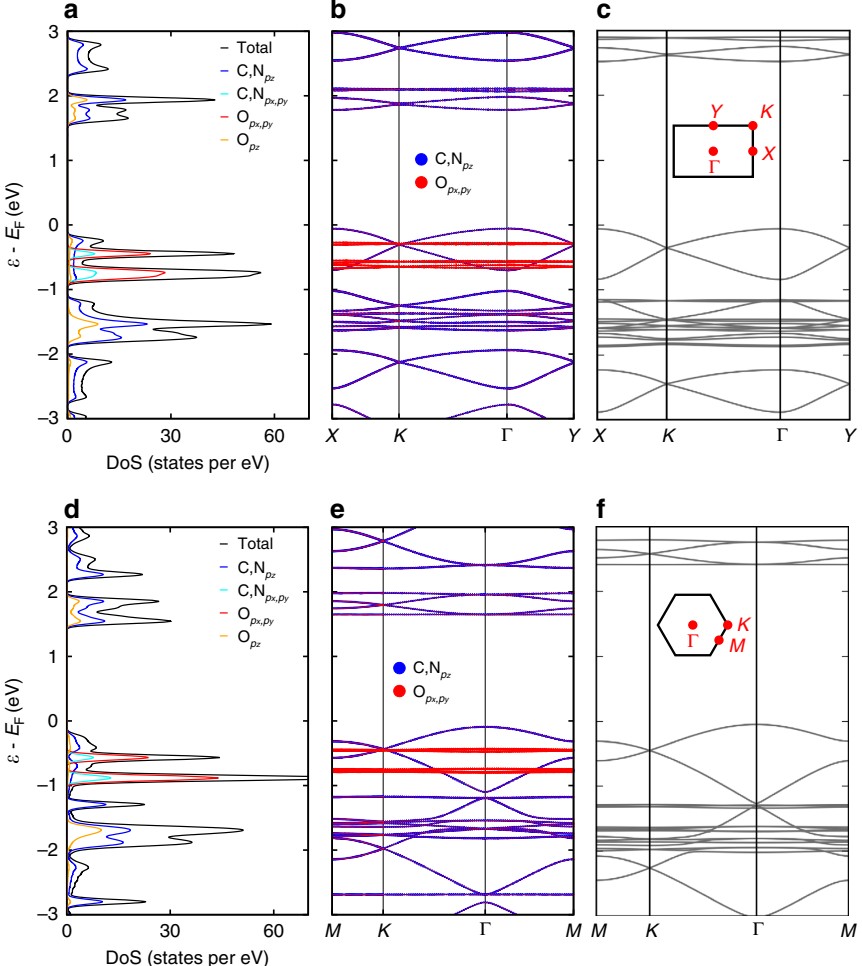

**Figure 6 | Electronic structure of the 1D chain and 2D polymer in vacuum.** Computed DoS (**a**) and band structure (**b,c**) of the 1D self-assembled CTPA chain at PBE + D3 and HSE level, respectively. DoS (**d**) and the band structure of 2D CTPA polymer at PBE + D3 (**e**) and HSE (**f**) level. The Brillouin zones and the relevant high symmetry points of 1D and 2D system are sketched in **c,f**.

| **Table 1 | Overview of computed and experimental band gaps.** | | | |
|---|---|---|---|
| | **DFT** | | **Experiment** |
| | **PBE + D3 (eV)** | **HSE (eV)** | $E_{\text{VBE} - \text{CBE}}$ **(eV)** | $E_{\text{VBE} - \text{CB}}$ **(eV)** |
| CTPA monomer | 2.27 | 3.29 | 3.91 ± 0.03 | |
| 1D (isolated) | 1.86 | 2.61 | 2.50 ± 0.09 | 3.31 ± 0.03 |
| 1D (self-assembled) | 1.84 (1.78) | 2.58 | | 3.18 ± 0.06 |
| 2D | 1.74 (1.57) | 2.46 | 2.45 ± 0.09 | 3.14 ± 0.04 |

1D, one-dimensional; 2D, two-dimensional; CB, conduction band; CBE, conduction band edge; CTPA, carbonyl-bridged triphenylamine; DFT, density functional theory; HOMO, highest occupied molecular orbital; LUMO, lowest unoccupied molecular orbital; PBE, Perdew–Burke–Ernzerhof; STS, scanning tunnelling spectroscopy; VBE, valence band edge.
Computed PBE and HSE band gaps in vacuum are compared with experimental results for 0D − 2D CTPA. For PBE, data for surface-supported structures are given in brackets. The HOMO–LUMO gap of CTPA **1** monomer, the $E_{\text{VBE} - \text{CBE}}$ and $E_{\text{VBE} - \text{CB}}$ are derived from STS experiments (see Fig. 4).

the structure, with nodal planes towards the nitrogen centre, the oxygen moieties and within the benzene units that connect the CTPA monomers to the polymer. This is in line with the weakly distinct band dispersion and the numerous flat bands. CB and CB + 1 are very similar, in line with the fact that they are observed together in STS experiments. No charge is located at nitrogen centres, whereas increased density is observed at the carbon atoms located at the edges of the newly formed C–C bonds. These features perfectly agree with the experimental CB density map in Fig. 5c. Last, CB + 2 again

shows no noticeable charge localized at the nitrogen centre but unique features on top of the newly formed C–C bonds. It is noteworthy that we also investigated the partial charge densities of states for the structures in vacuum (see Supplementary Fig. 13). The visible features remain the same, despite small shifts in the relative energies. This is true for both 2D and 1D structures, that is, the dI/dV maps of the isolated 1D chains (Fig. 5f–h), which are similar to the dI/dV maps in experiments, are explained by analogous features as discussed for the 2D network above.

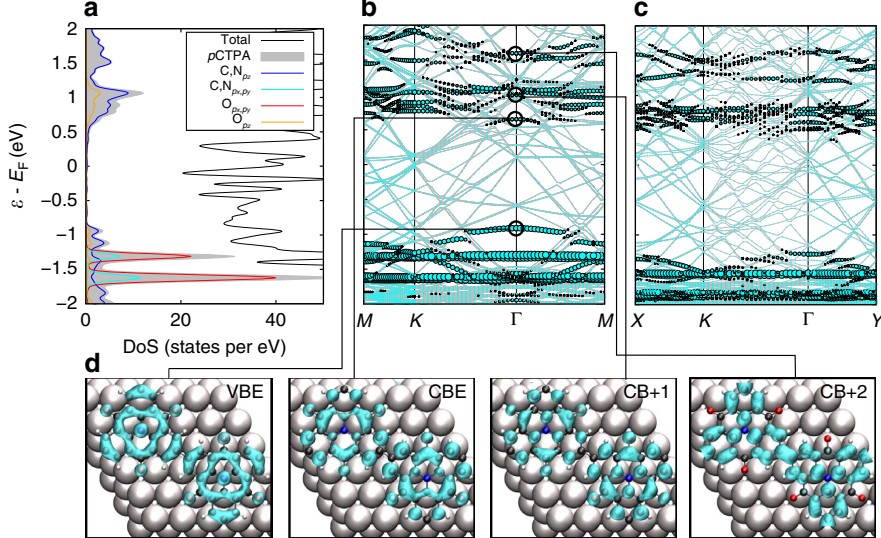

**Figure 7 | Electronic structure of the 1D chain and 2D polymer on Au(111).** (**a**) Projected DoS for the 2D polymer (pCTPA) adsorbed on Au(111). The adsorbate contributions are colored analogously to Fig. 6d. (**b**) The according band structure, with adsorbate bands coloured collectively cyan for better visibility. In addition, 3D representations of the projected DoS for the valence band (VB) and the first three CBs are shown at the Γ point for an isodensity value of 0.005 eÅ$^{-3}$ in **d**. (**c**) The analogous band structure of a 1D self-assembled linear chain on Au(111).

## Discussion

The hierarchical approach presented herein provides for structural control in the on-surface synthesis of functionalized porous 2D polymers that is less dependent on the growth kinetics. Using brominated macrocycles and chains as reaction intermediates constrains the defects to the rim of the formed 2D networks, leaving the properties of the covalent polymer largely preserved. The self-assembly of intermediate macrocycles and chains is essential to achieve structurally well-ordered covalent 2D networks: first, undesired structural elements segregate to the periphery of the network. Second, the weak, non-covalent interactions among the intermediates make them structurally controllable and lead to equilibrated structures. Covalent linking of macrocycles form defect-free 2D networks after the second reaction step, while chains are still susceptible to form pentagonal and heptagonal defects. Inevitably, 2D structures synthesized by Ullmann-type reactions are size limited due to the presence of halogens[46] on the surface and the reaction conditions that reduce the diffusion of the building blocks[16,47].

The hierarchical approach produces structurally related planar 1D chains and 2D networks that permits direct insight into the interplay between dimensionality and effective conjugation on the band gap of conjugated polymers. The observed decrease of the band gap with increased dimensionality is well known for inorganic nanostructures[48,49]; however, to our knowledge, it is shown here experimentally for the first time for organic molecular structures fabricated in a bottom-up approach. The decrease of the band gap and the relative shift of the frontier orbitals (stronger shift in the unoccupied states) is qualitatively in good agreement with DFT as shown here and for related dimethylmethylene-bridged triphenylamines[35]. The smaller reduction in the band gap energy from covalent 1D to 2D structures in comparison with the change between monomer and 1D polymer suggests that the delocalization length is increased to a smaller extent between 1D and 2D CTPA polymers. Our study clearly outlines the possibility of band gap engineering in 2D covalent organic structures. The functionalized pores provide further tunability of the band gap in future experiments and are interesting for potential applications in electronics, sensor devices and catalysis. The synthesis of triphenylamine derivatives is in this respective versatile, providing a variety of substitution possibilities to explore the tunability of band gap and host–guest interactions[50].

In summary, we disclosed the on-surface synthesis of macrocycles, 1D chains and 2D polymers with functionalized pores on Au(111) using Ullmann-type coupling reactions between halogen-substituted CTPAs. The high-resolution capability of STM to resolve the geometric and local electronic structure, in combination with DFT calculations, provides valuable details on the growth mechanism and band structure of functionalized 1D and 2D polymers on metal surfaces. First, the synthesis of the twofold symmetric precursors yields well-ordered self-assemblies of six-membered macrocycles and chains. The preferred hexagonal shape of the macrocycles results from the weak interaction forces between the macrocycles. Second, the hierarchical two-step synthesis using a CTPA precursor with a tuned substitutional pattern facilitates the fabrication of structurally well-defined 2D polymers consisting of prevalent six-membered pores with five- and seven-membered defects located solely at the periphery of the islands. Third, the covalent character of the 1D and 2D polymers is clearly identified in the d$I$/d$V$ and DoS maps: the valence band reveals an extended conjugation of the π-electrons along the molecule backbone and localized unfilled states of the CBs are found at the newly formed C–C bond. The electronic band gap was decreased by the increased dimensionality changing from the monomers to structurally related chains and network. The change in the gap is comparably larger from the monomer to the 1D polymer than from the 1D to the 2D system. Last, the functionalized pores show a remarkable combination of structural, chemical and electronic properties, which makes them particularly attractive for host–guest chemistry. The respective studies are currently being pursued in our laboratories and will be published elsewhere.

## Methods

**STM measurements.** All STM experiments were conducted with a low-temperature STM/ncAFM (non-contact atomic force microscope) from Scienta Omicron GmbH operated in UHV at a base pressure below $1 \times 10^{-10}$ mbar. The STM measurements were acquired in constant-current mode and at 4.5 K, if not indicated otherwise. The sample is grounded in our experiment; nonetheless, the

bias voltages mentioned refer to the sample. A mechanically cut Pt/Ir tip (90% Pt and 10% Ir) prepared by field-emission and controlled indentation in the Au(111) surface was used for topographic and spectroscopic measurements. The d$I$/d$V$ spectra and maps, shown in Figs 4 and 5, were measured by lock-in technique with a frequency of 721.987 Hz and modulation amplitude of 12 mV (root mean square). The functionalized tip in Fig. 2c,e was obtained by controlled pulsing and indentation over a molecular layer. The STM images were analysed using the WSxM software[51]. The STS spectra in Fig. 4 were normalized and smoothed.

**Sample preparation.** A clean Au(111) was prepared by repeated cycles of Ar$^-$ sputtering (1 kV) followed by annealing at 700 K. Single crystals (Mateck) and Au thin films on mica (Phasis) were used. The molecules were evaporated with a rate of 0.1 monolayers per minute from a commercial Knudsen cell (Kentax GmbH) with the crucible held at temperatures between 500 and 550 K for the triphenylamine precursor compounds. The rates were determined by a quartz crystal microbalance. The Au(111) substrate was kept at 300 K during the evaporation of the CTPA precursor **2** and **3**, and was subsequently heated to initiate the coupling reaction. CTPA compound **1** was deposited on Au(111) held at 80 K to obtain isolated monomeric species. The molecules were thoroughly degassed prior deposition on the surface.

**Theoretical calculations.** Closed shell DFT calculations were carried out with the Vienna *ab initio* simulation package[52] employing a plane wave basis set up to a kinetic energy threshold of 415 eV and the projector-augmented wave method[53] for the description of core electrons. The applied PBE exchange-correlation functional[54] was supplied with the D3 correction[55] (using Becke–Johnson damping[56]) to account for dispersive interactions. Energies and geometry optimizations were converged to 10$^{-6}$ eV and forces acting on ions below 0.001 eV Å$^{-1}$, respectively. Free-standing systems in vacuum were computed with 10 Å separating periodic mirror images into the $z$-direction. Calculations on gold were carried out using a $(6 \times 6)$ replica of an optimized Au(111) $(1 \times 1)$ slab containing three layers, that is, we are neglecting the reconstruction of the Au(111) surface that occurs in experiments. This is solely done, as the latter is impractical in calculations and not necessary for a weakly interacting surface at the given level of theory, as proven by the good agreement with experimental results. Therefore, we also did not further investigate the structural alignments of the adsorbate with respect to the substrate in detail. In fact, two different tested structures with the central N atoms over Au(111) fcc-hollow and top sites yield almost isoenergetic adsorption energies (the latter being favored by 0.05 eV per CTPA unit) with identical adsorption distance. During optimizations, only the topmost layer was allowed to relax, while the bottom two were fixed at their bulk positions. Band structures were evaluated on these optimized structures using PBE and the HSE06 hybrid functional[57]. Owing to the metallic character of the system, a first order Methfessel–Paxton level broadening[58] with half-width of $\sigma = 0.15$ eV was used. To account for the finite size of the slab model, a dipole correction[59] was employed into $z$-direction. Reciprocal space was sampled using a $3 \times 3 \times 1$ Monkhorst–Pack grid[60]. DoS were computed with a decreased level broadening and an increased **k** point grid ($\sigma = 0.05$ eV and $9 \times 9 \times 1$ **k** points). Adsorption energies are defined throughout as $E_{ads} = E(X/Au(111)) - E(X) - E(Au(111))$, that is, subtracting the energies of the isolated systems from the combined one. Charge-density differences were computed alike by subtracting respective charge densities $\varrho^{CDD} = \varrho(X/Au(111)) - \varrho(X) - \varrho(Au(111))$. For band structures we carried out calculations employing the HSE exchange-correlation functional[61] in addition to those using the PBE functional.

**Synthesis of precursor molecule.** The CTPA compounds **1**, **2** and **3** were synthesized from known $O,O',O''$-amino-trisbenzoic acid-trimethylester (**S1**)[62], upon halogenation with I$_2$ in the presence Ag$_2$SO$_4$ and $N$-bromosuccinimide, respectively, followed by saponification and Lewis acid-catalysed Friedel–Crafts acylation of the *in situ* formed acid chloride. For experimental details and characterization data, see Supplementary Methods and Supplementary Fig. 14.

**Data availability.** The data that support the findings of this study are available from the corresponding author on request.

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

## Acknowledgements

This work was supported by the German Research Foundation (DFG) through the Collaborative Research Center SFB 953 'Synthetic Carbon Allotropes', GRK 1896 'In-Situ Microscopy with Electrons, X-rays and Scanning Probes', the priority program SPP 1807 'Control of London Dispersion in Molecular Chemistry' and the Cluster of Excellence EXC 315 'Engineering of Advanced Materials' at the Friedrich-Alexander University Erlangen-Nürnberg and the ERC Starting Grant SURFLINK (contract number 637831). M.K. thanks the 'Solar Technologies Go Hybrid' initiative of the Free State of Bavaria and the German Fonds der Chemischen Industrie for their generous support. J.G. thanks the German Research Foundation for support from Research Fellowship GE 2827/1-1.

## Author contributions

C.S., M.A., Z.Y. and S.M. conducted the STM/STS measurements and analysed the STM/STS data. A.H., N.H. and M.K. synthesized the precursor molecules. J.G. and A.G. carried out the DFT calculations. S.M., M.K. and J.G. wrote the manuscript. S.M. and M.K. conceived the project.

## Additional information

**Competing financial interests:** The authors declare no competing financial interests.

