## [Peer Review File · Nature Communications]

Reviewers' comments:

Reviewer #1 (Remarks to the Author):

The manuscript by Maier et al. reports on the on-surface synthesis of 1D and 2D polymers in ultra-high vacuum. A halogenated triphenylamine precursor molecule is evaporated onto a Au(111) surface and subsequently forced to polymerize through a Ullmann-type coupling scheme by applying thermal energy. Scanning tunneling microscopy (STM) is used to characterize the topography of the polymeric structures and scanning tunneling spectroscopy (STS) allows measuring the electronic frontier orbitals of the polymers. Experimental findings are supported by density functional theory calculations (DFT).

The paper addresses the changes in electronic structure comparing 1D with 2D polymers. So far, previous theoretical studies have suggested that the electronic structure indeed depends on the dimension of the polymer and a smaller HOMO-LUMO gap for 2D polymers compared to 1D is predicted. The present work shows for the first time at the single polymer level that the predictions are correct and that the gap in this particular 2D polymer is 0.2 eV smaller than in the corresponding isolated 1D polymer build up from the same structural repeat unit.

The experimental work seems to be well carried out and the DFT calculations carefully crafted.

I suggest publication of the manuscript if following points can be addressed by the authors:

Fig. 2 b,d and d,e are mixed in caption, also in the main text.

The H-bonds that stabilize the two self-assembled structures (macrocycles and 1D polymers) are not described well in the manuscript. A detailed model including discussion (bonding between what atoms, bond lengths, etc.) should be included at least in the supporting information. The existing DFT calculations on the 1D polymer self-assembly should give additional relevant information.

The term 'porous graphene' (page 7, line 140 and thereafter) is commonly used for the structure synthesized by Bieri, Fasel et al. (doi: 10.1039/B915190G). To avoid confusion, I suggest relabeling the triphenylamine 2D polymer.

Certain emphasis is given to the functionalized pores of the network in the manuscript. The authors might like to point out that a 2D polymer with functionalized pores is an established concept (Porte et al. doi: 10.1039/C3CE26494G, Lei et al. doi: 10.1039/C5CC02232K)

The measured band gap depicted in Fig. 4 is roughly twice as large as the calculated gap of Fig. 6. This discrepancy should be discussed and it should be pointed out that the PBE functional severely underestimates band gaps. Why was it used then? How do the experimental gaps compare to the very similar polymer calculated by Gutzler & Perepichka (doi: 10.1021/ja408355p), who used the arguably more suitable B3LYP functional?

Starting on page 13, the gap reduction from free-standing to adsorbed 2D polymer is discussed (based on DFT). To exclude that the observed energy shift in the LUMOs of 2D polymer referenced to single strand 1D polymer is not a consequence of a substrate interaction, DFT calculations on a single 1D polymer including substrate should also be provided.

In Fig. 7b the electron density is very hard to distinguish from the Au background. This figure should be improved.

Fig. 6 needs Brillouin zones, in particular for the self-assembled 1D case (is X supposed to be Y in Fig. 6b, abscissa?, see also line 259, page 12).

It is noteworthy that band dispersion is as strong in between 1D polymers as for the covalent structure (2D), see band plots. The origin is roughly discussed in the manuscript, but what is the advantage then in using covalent polymers if their synthesis is so complicated compared to self-assembled structures?

Page 12, lines 240-241: 'The presence of unoccupied states at the positions of the newly formed C-C bond is a clear signature for the covalent nature of the intermolecular bond.'

This is not necessarily always true, also other bond types can have bright contrast, e.g. organometallic bonds, where a metal atom sits coordinated between two dehalogenated molecules.

Reviewer #2 (Remarks to the Author):

Steiner et al. report on the on-surface synthesis and the structural and electronic structure properties of macrocycles, 1D chains, and 2D polymers with functionalized pores. The findings are based on Ullmann-type coupling reactions and low-temperature scanning tunneling microscopy measurements that are supported by first-principles density functional theory calculations. The used methodologies are appropriate and the conclusion is robust. Their manuscript offers a thoughtful and well-written analysis of the results and provides a very interesting and pioneering experimental perspective on the dependence of micro-porous covalent nanostructures' properties on dimensionality. After revisions as noted below, this paper would be acceptable for publication in Nature Communications.

The structure of the polymer raises a few questions. Why was carbonyl-bridged triphenylamines (CTPAs) chosen for the investigation? While a noticeable bandgap reduction was observed from oligomer to 1D chain, very small bandgap difference between 1D and 2D polymers was found and ascribed by the authors to a less effective conjugation. It would be more interesting for 2D materials if the authors could use other precursors to fabricate 2D polymers with remarkable bandgap differences from 1D ones in the future.

As can be seen from STM images, there are noticeable edges and defects in the polymeric systems. The authors are suggested to present some discussions on the edge or defect effect for the material's electronic structure by additional experiments or calculations.

Point-by-point Response to NCOMMS-16-19472

Reviewer 1:

The manuscript by Maier et al. reports on the on-surface synthesis of 1D and 2D polymers in ultra-high vacuum. A halogenated triphenylamine precursor molecule is evaporated onto a Au(111) surface and subsequently forced to polymerize through a Ullmann-type coupling scheme by applying thermal energy. Scanning tunneling microscopy (STM) is used to characterize the topography of the polymeric structures and scanning tunneling spectroscopy (STS) allows measuring the electronic frontier orbitals of the polymers. Experimental findings are supported by density functional theory calculations (DFT).

The paper addresses the changes in electronic structure comparing 1D with 2D polymers. So far, previous theoretical studies have suggested that the electronic structure indeed depends on the dimension of the polymer and a smaller HOMO-LUMO gap for 2D polymers compared to 1D is predicted. The present work shows for the first time at the single polymer level that the predictions are correct and that the gap in this particular 2D polymer is 0.2 eV smaller than in the corresponding isolated 1D polymer build up from the same structural repeat unit.

The experimental work seems to be well carried out and the DFT calculations carefully crafted.

I suggest publication of the manuscript if following points can be addressed by the authors:

Author reply: We appreciate the positive feedback and provide in the following a detailed point-by-point response to the raised questions.

Fig. 2 b,d and d,e are mixed in caption, also in the main text.

Author reply: The caption of Fig. 2 has been fixed and also the main text has been corrected correspondingly.

The H-bonds that stabilize the two self-assembled structures (macrocycles and 1D polymers) are not described well in the manuscript. A detailed model including discussion (bonding between what atoms, bond lengths, etc.) should be included at least in the supporting information. The existing DFT calculations on the 1D polymer self-assembly should give additional relevant information.

Author reply: We provided additional information on the bonding motifs of the 1D polymer self-assembly in the Supporting Information (Fig. 1). Bond lengths from the DFT calculations are included. In addition, we included the DFT model of the CTPA chains on Au(111) in Fig. 2. DFT calculations on the self-assembly of the macrocycles have currently not been performed.

The term 'porous graphene' (page 7, line 140 and thereafter) is commonly used for the structure synthesized by Bieri, Fasel et al. (doi: 10.1039/B915190G). To avoid confusion, I suggest relabeling the triphenylamine 2D polymer.

Author reply: Porous graphene generally refers to a graphene-like structure exhibiting nanometer-sized pores, and Bieri et al. successfully synthesized 2D polyphenylene as a prototype of porous graphene. Because our triphenylamine 2D polymer has a strongly planar structure we consider the nomenclature "porous graphene-like" networks as appropriate.

Certain emphasis is given to the functionalized pores of the network in the manuscript. The authors might like to point out that a 2D polymer with functionalized pores is an established concept (Porte et al. doi: 10.1039/C3CE26494G, Lei et al. doi: 10.1039/C5CC02232K)

Author reply: Many thanks to the reviewer. The suggested references have been added to the introduction (new ref 28 and 29).

The measured band gap depicted in Fig. 4 is roughly twice as large as the calculated gap of Fig. 6. This discrepancy should be discussed and it should be pointed out that the PBE functional severely underestimates band gaps. Why was it used then? How do the experimental gaps compare to the very similar polymer calculated by Gutzler & Perepichka (doi: 10.1021/ja408355p), who used the arguably more suitable B3LYP functional?

Author reply: In this study, we decided to not use B3LYP calculations, since they are not suitable for periodic structures. We clearly state that DFT underestimates band gaps on p.12 and emphasize that only qualitatively correct trends can be expected. The latter can be trusted with a high level of confidence for the studied elements and material class. Nevertheless, we performed additional new calculations for all four structures (monomer, 1D chain isolated/self-assembled and 2D network) using a (long range) screened hybrid functional, the HSE06 functional, which is better suited for the study of periodic systems than B3LYP.

For the gas phase the HSE calculations show the expected results: i) the band gaps are systematically increased, being now in good agreement with our STS experiments. We included in the manuscript Table 1, which provides a direct comparison between the computed band gaps and the experimental STS results. ii) differences in the band structure, e.g., the down shift of the oxygen bands in the valence band region, is alike for all structures, and thus iii) relative trends on the band gap decrease from 0D to 2D structures are described correctly with standard DFT. Therefore, we are confident that the obtained results in gas phase (and the discussed differences between both methods) are transferable to the systems that include the substrate, where no hybrid-functional treatment is possible due to the increased computational cost.

Starting on page 13, the gap reduction from free-standing to adsorbed 2D polymer is discussed (based on DFT). To exclude that the observed energy shift in the LUMOs of 2D polymer referenced to single strand 1D polymer is not a consequence of a substrate interaction, DFT calculations on a single 1D polymer including substrate should also be provided.

Author reply: It is unlikely that the 1D chains interact differently with the substrate than the 2D polymers, especially, since the DFT calculations demonstrate physisorption of the polymer with an adsorption distance of 3 Å above the surface, i.e., the interaction between the polymer and the surface is small and solely based on distance-dependent electrostatic effects (see the supporting information for more details). However, we now provide new results of 1D chains on the Au substrate (structural model: Fig. 2; band structure Fig. 7). Note that we could not model an isolated 1D chain, since such a model would require an impedingly large Au surface. The new results demonstrate that the effect of dimensionality is retained and that the substrate has a similar effect on both 1D and 2D systems.

In Fig. 7b the electron density is very hard to distinguish from the Au background. This figure should be improved.

Author reply: We improved the figure.

Fig. 6 needs Brillouin zones, in particular for the self-assembled 1D case (is X supposed to be Y in Fig. 6b, abscissa?, see also line 259, page 12).

Author reply: Sketches of the Brillouin zones together with the relevant high symmetry points are now included into Figure 6.

It is noteworthy that band dispersion is as strong in between 1D polymers as for the covalent structure (2D), see band plots. The origin is roughly discussed in the manuscript, but what is the advantage then in using covalent polymers if their synthesis is so complicated compared to self-assembled structures?

Author reply: The 2D porous polymers provide the opportunity to further tune the electronic properties through coadsorption of molecules/atoms in the pores. In contrast the self-assembled

1D polymer exhibits a closed-packed structure. Further, the porous 2D networks provide a high structural stability, which is usually not achieved in self-assembled structures.

Page 12, lines 240-241: 'The presence of unoccupied states at the positions of the newly formed C-C bond is a clear signature for the covalent nature of the intermolecular bond.'

This is not necessarily always true, also other bond types can have bright contrast, e.g. organometallic bonds, where a metal atom sits coordinated between two dehalogenated molecules.

Author reply: We agree with the reviewer that this sentence is misleading and changed it accordingly to: 'The presence of unoccupied states at the positions of the newly formed C-C bond has also been observed in previous STS experiments on covalent networks.'

Reviewer 2:

Steiner et al. report on the on-surface synthesis and the structural and electronic structure properties of macrocycles, 1D chains, and 2D polymers with functionalized pores. The findings are based on Ullmann-type coupling reactions and low-temperature scanning tunneling microscopy measurements that are supported by first-principles density functional theory calculations. The used methodologies are appropriate and the conclusion is robust. Their manuscript offers a thoughtful and well-written analysis of the results and provides a very interesting and pioneering experimental perspective on the dependence of micro-porous covalent nanostructures' properties on dimensionality. After revisions as noted below, this paper would be acceptable for publication in Nature Communications.

The structure of the polymer raises a few questions. Why was carbonyl-bridged triphenylamines (CTPAs) chosen for the investigation? While a noticeable bandgap reduction was observed from oligomer to 1D chain, very small bandgap difference between 1D and 2D polymers was found and ascribed by the authors to a less effective conjugation. It would be more interesting for 2D materials if the authors could use other precursors to fabricate 2D polymers with remarkable bandgap differences from 1D ones in the future.

Author reply: The carbonyl-bridged triphenylamines (CTPA), were used for a couple of reasons, as detailed through the text:

- The interesting opto-electronic and redox properties of the family of TPAs (p.5)
- The rigidity and 3-fold symmetry of the precursor (p.7)
- The possibility to attach functional groups pointing towards the pores (p.3)

We agree with the review that the band gap difference between 1D and 2D polymers is comparably small. To further elaborate on the magnitude of the band gap decrease as a result of extending π -conjugation in the polymeric structures, we performed additional STS experiments on the CTPA monomer for comparison. We found that the gap decreases from the monomer to the 1D chains much stronger, which indicates in fact quite an efficient conjugation among the building blocks. The difference between the energy gaps of 1D and 2D structures is less pronounced than between 0D and 1D. This means that the delocalization length is hardly increased between 1D and 2D CTPA polymers.

As can be seen from STM images, there are noticeable edges and defects in the polymeric systems. The authors are suggested to present some discussions on the edge or defect effect for the material's electronic structure by additional experiments or calculations.

Author reply: We thank the referee for the interesting remark on defects and edge effects. This manuscript focuses on the band structure of defect-free polymeric structures with different dimensionality. We took great care to avoid structural defects and impurities (Br, I) in the spectroscopic data to ensure excellent reproducibility. We agree that a detailed spectroscopic analysis of defects and edge structures would be very interesting, however beyond the scope of this manuscript.

Concerning edge effects we note: The dI/dV maps, performed on the networks generally do not show any pronounced contrast difference between CTPA units at the edge of the network or within the network. The same is the case for the end caps of wires.

Summary of changes:

In addition to the changes mentioned above, the following changes were added in the resubmitted manuscript:

- The abstract is modified in length and structure to meet the requirements of the Journal (150 words max., 2-3 introducing sentences, major results, description of conclusion).
- Fig. 1: Monomer is added and the CTPA compounds were relabeled throughout the manuscript.
- Fig. 2: DFT model of the chain on Au(111) was added in (f) and described on page 6.
- Fig. 3: DFT model was updated to enhance the visibility of the CTPA polymer.
- Fig. 4: CBE for the 2D network and free-standing chain were included with a corresponding discussion on page 10.
- Fig. 5: STS maps of the monomer were included.
- Fig. 6: HSE band structures were included as well as sketches of the Brillouin zone with a corresponding discussion on page 13-14.
- Table 1 was added to compare the band gaps calculated by the different functional.
- Fig. 7: Band structure of the chains on Au(111) has been added in (c) and layout of (d) was improved according to the request of reviewer 1. A corresponding discussion to the band structure of the chains on Au(111) was included on page 15-16
- "Fig. S" was replaced by "Supplementary Fig." throughout the manuscript and Supporting information.
- Fig. S1: DFT model of the chain in gas phase and on the Au(111) surface was added.
- Fig. S5: STS measurements of the monomer 1 were added.
- Fig. S8: STM and STS measurements of the monomer 1 were added.
- Fig. S9: STM images of the monomer 1 were added.
- Fig. S11 has been included to compare the HSE calculated band structure of the free-standing CTPA polymer with different dimensionalities.
- Small improvements of the language that do not change the conclusions of the paper were added in the resubmitted manuscript.

REVIEWERS' COMMENTS:

Reviewer #1 (Remarks to the Author):

The changes to the manuscript improved the quality of the paper considerably, and reviewer's remarks have been thoroughly addressed.
The paper should be published in the current form.

Reviewer #2 (Remarks to the Author):

The authors have thoroughly addressed my previous comments and concerns. I recommend publication of this manuscript without delay.

Response to NCOMMS-16-19472A

Reviewer #1 (Remarks to the Author):

The changes to the manuscript improved the quality of the paper considerably, and reviewer's remarks have been thoroughly addressed.

The paper should be published in the current form.

Author reply: We thank Reviewer #1 for supporting our work.

Reviewer #2 (Remarks to the Author):

The authors have thoroughly addressed my previous comments and concerns. I recommend publication of this manuscript without delay.

Author reply: We thank Reviewer #2 for supporting our work.